# Weakly-Supervised Semantic Segmentation via Transformer Explainability

## Reproducibility Summary

**Scope of Reproducibility**

In this work, we experimented with Layer-wise Relevance Propagation and combined it with back-propagation to perform classification and semantic segmentation, following the approach proposed by Chefer H. et al., in (1) for computer vision. Moreover, we incorporated the concept of pixel affinities, by using ViT-based explainability as visual seeds to drive the generation of pseudo segmentation masks by computing pixel affinities, following the approach described by Ahn J. et al. in (2).

**Methodology**

In order to reproduce the experiments presented in (1) and (2), we initially examined the authors' code thoroughly and based on our understanding, we tried to replicate most parts of the pipeline apart from evaluation metrics for positive and negative perturbation area-under-curve (AUC) results for the predicted and target classes on the ImageNet (3) validation set, as well as Segmentation performance on the ImageNet-segmentation (4) dataset, which we borrowed from the authors' repository for the work of Chefer H. et al., in (1). Regarding hardware, we used private resources to train our ViT-hybrid architecture and Affinity network, as well as perform inference for all our models; Finally, it took roughly 15 GPU hours to reproduce the vision-related results of (1) whereas it took about 40 GPU hours to train and evaluate the AffinityNet on the Hybrid-ViT architecture.

**Results**

Overall, we reproduced the experiments related to the vision task as conducted at (1). Our results are up to first decimal place identical to those reported in (1) thus supporting the authors' claim of having implemented a relatively sufficient ViT interpretability method. When it comes to the AffinityNet (2), the method has been adapted in the context of Hybrid-ViT architectures with our experiments indicating that the weakly-supervised semantic segmentation performance of Hybrid-ViT architectures are inferior to the CNN-based ones.

**What was easy**

We found particularly easy to run and understand the code provided by the original authors of both (1) and (2) papers. When it comes to replicating (1), the authors provided most of the information required to reproduce the vision-related experiments with the code compensating for what was missing.

**What was difficult**

The main difficulty of replicating the study presented in (1) was that details on how to compute the AUC metric were not provided in the paper report.

# 1 Introduction

One of the most exciting technological aspects nowadays is Machine Learning's mind-blowing potential in transforming the world we live in, mainly due to its exciting resurgence through Deep Learning. However, as machine learning models become more complex, there is a noticeable trade-off between accuracy and simplicity or interpretability (5) and plenty of cutting-edge research papers have been published in top-tier conferences related to this tension. In this project, we primarily experimented with Layer-wise Relevance Propagation (LRP), a mechanism of explaining what pixels are relevant within a 2-dimensional image for reaching a classification decision (6) and applied it to a Vision Transformer [ViT] (7), combined with gradient back-propagation to perform classification but also semantic segmentation on the respective data in ImageNet (3; 4), by reproducing the work of Chefer H. et al, in (1).

Furthermore, the task of semantic segmentation refers to clustering the pixels of an input image that correspond to the same semantic category. There are various approaches dedicated to this task with the one proposed in (8) being the current state-of-the-art. However, they all rely on training given ground truth segmentation masks. Considering that annotating images in the form of segmentation masks is a rather expensive and tedious process, capitalizing on weak forms of segmentation would be highly beneficial. In order to address these issues, in this project, we investigated using ViT-based explainability as visual seeds to drive the generation of pseudo segmentation masks by computing pixel affinities, following the approach described in (2). In particular, we trained a Hybrid ViT-base, where the patches are extracted from a CNN feature map, through relevance propagation and used those as seeds to a network computing pixel affinities, in order to improve quality of the generated segmentation masks.

# 2 Related Work

Semantic segmentation has numerous applications, such as self-driving cars or medical image analysis. Additionally, the evident importance in providing the machines with the ability to perceive the world along with its challenging nature has attracted many researchers to this domain. Many algorithms have been proposed for this task with Mask R-CNN (9) being among the most frequently employed ones. Although such approaches can be trained to extract semantic with high precision, they require an extensive amount of semantically annotated training samples. In their work (2), the authors capitalize on image-level supervision to construct competent pseudo-segmentation masks that can be further utilized to train the segmentation approaches requiring ground truth labels. More specifically, they use class activation mapping (CAM) (10) seeds to model the relation between neighboring pixels, which enables the refinement of the initial CAM cues into segmentation masks of higher quality. Although the previous approach results in relatively accurate segmentation masks, the initial CAMs seeds tend to highlight only the most descriptive part of an instance, which negatively affects the quality of the generated segmentation masks. With the purpose of mitigating this issue, the essayist of (11) employs a sub-category exploration approach.

Regarding Deep Neural Networks (DNNs) interpretability, various approaches have been proposed in the literature. GradCAM (12) is a popular interpretability method applied to various CNN architectures that weighs feature activations in different pixel regions within an image with the average gradient of the class scores. After these gradients are computed through global average pooling, they are passed to a ReLU[1] activation function that intensifies pixels contributing towards increasing the target class activation scores. However GradCAM is restricted to CNN architectures. One more general approach is RISE (13) that measures pixels' importance by applying element-wise multiplications of the original input with a sampled random binary mask to reduce their intensities to zero and only preserve the most important among them.

Although CNN-based architectures have demonstrated competent performance in a number of vision-related tasks, they come with an increased inductive bias due to the 2D neighboring structure of the

---

[1]Rectified Linear Units activation function is: $\text{ReLU}(x) = \max\{x, 0\}$.

| Model | Layers | Hidden size $D$ | MLP size | Heads | Params |
|---|---|---|---|---|---|
| ViT-Base | 12 | 768 | 3072 | 12 | 86M |

Table 1: Details of ViT model variants. Table extracted from (7).

images. On the other hand, transformer-based architectures are able to learn spatial relationships detached from the explicit 2D nature of the images. Transformer architecture, since it was proposed in 2017 by Waswani A. et al., (14) has become very popular in various deep learning domains, and it is based solely on attention mechanisms, dispensing recurrence and convolutions entirely and weighing the influence of different parts of the input data. Following its recent success in NLP, it was recently adopted in computer vision tasks, and in this work, we focus on particularly re-implementing a Vision Transformer [ViT] (7) from scratch. Additionally, we employ the explainability cues derived from a image classification ViT to drive the construction of segmentation masks given solely image-level annotation as we explain hereunder.

## 3 Methods

In this section, we describe the methods utilized in our work. Precisely, in subsection 3.1, we provide details about Vision Transformer architecture. Subsection 3.2 explains how we perform relevance propagation in our model implementations. Finally, in subsection 3.3, we present the AffinityNet framework modeling the affinity of neighboring pixels.

### 3.1 ViT Classification

As mentioned earlier, a Vision Transformer [ViT] (7) is an implementation of transformer networks for computer vision tasks. The transformer encoders in ViT are similar to the original transformer architecture introduced in (14) with slight modifications in the order of operations. Similarly to how a sentence is split into tokens, in ViT we split an image into patches and provide the linearization of the patches representations as input to stacked transformer encoders after adding positional embeddings. Positional embeddings are learned during training; while processing the input patches in given order $x_0, x_1, x_2, ...$ we learn the respective positional embeddings $\hat{x}_0, \hat{x}_1, \hat{x}_2, ...$ for the patches and compute the loss in a backward fashion. The input is then propagated to the attention heads, where multi-head attention is calculated as the concatenation of self-attention scores computed in each head individually as stated in the formulas below:

$$\text{Attention}(Q, K, V) = \text{softmax}\left(\frac{QK^T}{\sqrt{d_k}}\right) V$$

$$\text{Multihead}(Q, K, V) = \text{Concat}(\text{head}_1, ...\text{head}_h)\Theta^o$$

$$\text{where head}_i = \text{Attention}(Q\Theta_i^Q, K\Theta_i^K, V\Theta_i^V)$$

Attention is a mechanism for weighting representations learned in a neural network. It is proportional to the respective weights of the network and really flourished within a variety of NLP tasks, where self-attention and multi-head attention became one of the major breakthroughs in sequence modeling tasks precisely (15). In our implementation, we use ViT-Base, the smallest ViT model variant, which consists of 12 stacked encoder layers, as well as 12 attention heads in every layer, as it is illustrated in table 1. We use a [CLS] learnable embedding $\mathbf{z}_0^0 = \mathbf{x}_{\text{class}}$ to the sequence of embedded patches, whose state at the output of the Transformer encoder $\mathbf{z}_0^L$, to which a classification head is attached to represent an image $\mathbf{y} = \text{LayerNorm}(\mathbf{z}_0^L)$. We also employ a hybrid architecture, which again consists of a ViT-Base but the patches are extracted from a CNN feature map, while layer normalization is applied before every block and residual connections after every block in our implementation as it is described in (7).

## 3.2 ViT Explainability

As we explained in section 1, one of our main goals in this project was to apply LRP (6) to a ViT-Base model (7), combined with classic gradient back-propagation regime to perform classification but also semantic segmentation on the respective data in ImageNet (3; 4), by reproducing the work of Chefer H. et al, in (1). Considering the input feature map and weights of layer $n$ in form of tensors, $\mathbf{X}, \boldsymbol{\Psi}$ we compute the Deep Taylor Decomposition $R_j^{(n)}$ for relevance propagation as formulated below. This expression satisfies the conservation rule that broadly suggests that relevance will be maintained in consecutive layers.

$$R_j^{(n)} = \mathcal{G}\left(\mathbf{X}, \boldsymbol{\Psi}, R^{(n-1)}\right) = \sum_i \mathbf{X}_j \frac{\partial L_i^{(n)}(\mathbf{X}, \boldsymbol{\Psi})}{\partial \mathbf{X}_j}$$

Moreover, in cases we have two operators (e.g. skip connections and matrix multiplication) the above expression is used for both the input pairs $(u, v)$ and $(v, u)$ to compute $R_j^{u(n)}$ and $R_j^{v(n)}$. Given two such tensors $u$ and $v$, if we add them in layer $n$ the conservation rule is maintained but not in other cases of operations such as matrix multiplication. To address this lack of conservation we normalize the relevances and get $\bar{R}_j^{u(n)}$ and $\bar{R}_j^{v(n)}$ respectively. In addition, there is a special case related to the matrix multiplication operation, where we get two attribution maps for each of the matrices we multiply, and the sum of the relevances of each matrix equals $R$. Furthermore, to actually normalize the CAMs, all we need to do is divide each of them by 2, which is what the normalization below would do since $R_j^{u(n)}$ and $R_j^{v(n)}$ have identical sums.

$$R_j^{u(n)} = \mathcal{G}\left(u, v, R^{(n-1)}\right)$$
$$R_j^{v(n)} = \mathcal{G}\left(v, u, R^{(n-1)}\right)$$
$$\bar{R}_j^{u(n)} = R_j^{u(n)} \frac{\left|\sum_j R_j^{u(n)}\right|}{\left|\sum_j R_j^{u(n)}\right| + \left|\sum_k R_k^{v(n)}\right|} \cdot \frac{\sum_i R_i^{(n-1)}}{\sum_j R_j^{u(n)}}$$
$$\bar{R}_k^{v(n)} = R_k^{v(n)} \frac{\left|\sum_k R_k^{v(n)}\right|}{\left|\sum_j R_j^{u(n)}\right| + \left|\sum_k R_k^{v(n)}\right|} \cdot \frac{\sum_i R_i^{(n-1)}}{\sum_k R_k^{v(n)}}$$

Following the above formulas, we have computed relevances for all layers of our ViT-Base and have implemented relevance propagation, in order to perform semantic segmentation on the ImageNet-segmentation (4) dataset following the experiments described in (1). An example of a CAM generated by our Hybrid ViT-base, where the patches are extracted from a CNN feature map, through relevance propagation is illustrated in Fig. 1(b).

## 3.3 AffinityNet

At this stage, we employed the AffinityNet proposed in (2) with the purpose of refining the initially incomplete explainability cues, derived from the Hybrid-ViT network, into segmentation masks of higher quality. In more detail the AffinityNet aims at modelling the relation between adjacent pixels through leveraging the images' feature representation $f^{\text{aff}}$ and computing the similarity of $i^{\text{th}}$ and $j^{\text{th}}$ pixels as:

$$W_{i,j} = \exp\left(-||f_i^{\text{aff}} - f_j^{\text{aff}}||\right)$$

Conceptually, the AffinityNet is trained to predict the inter-pixel semantic affinities, in a class-agnostic manner, by learning to extract meaningful representations for each pixel. Evidently, target labels are required in order to drive the AffinityNet's weights towards accurately predicting the affinities.

### 3.3.1 Semantic Affinity Targets

Training the AffinityNet to model the inter-pixel relationships, requires supervision in the form of segmentation masks. In our scenario, ground truth segmentations labels were not provided and

thus the generated ViT explainability seeds are utilized as our best available source of supervision. Admittedly, the generated explainability cues can be quite incomplete and by no means precisely capture the whole instances, however, we can use the most confident pairs in terms of belonging to the same instance. Assuming $C$ classes with $M_c$ corresponding to the explainability cue of class $c$, we construct the background activation map $M_{bg}$ as:

$$M_{bg}(x,y) = \left[1 - \max_{c \in C} M_c(x,y)\right]^\alpha$$

The parameter $\alpha$ controls how confident the generated background cues are. Intuitively, when the $\alpha$ parameter is relatively high, a pixel of high activation in the $M_{bg}$ would be a strong indication of the pixel belonging to the background category. On the contrary, when the $\alpha$ parameter is relatively low, a high background activation suggests that background is the dominant semantic of that pixel but not with as much confidence. Next, we make use of the common practice of applying dense conditional random fields (dCRF) (16) to refine the activation responses for all $C + 1$ classes. Applying the dCRF on these classes' activations with the $M_{bg}$ having been derived from a low $\alpha$, favors classifying the pixels as background. On the other hand, when a high $\alpha$ is used, the dCRF is more prone to classifying a pixel as its most activated class. Having said that, applying dCRF on low $\alpha$ gives rise to the confident pixel of foreground instance while on the other, a high $\alpha$ allows for identifying confident background pixels. In our experiments, we set $\alpha_{\text{low}} = 4$ and $\alpha_{\text{high}} = 32$ respectively. Below we provide an indicative illustration of confident background and foreground pixels.

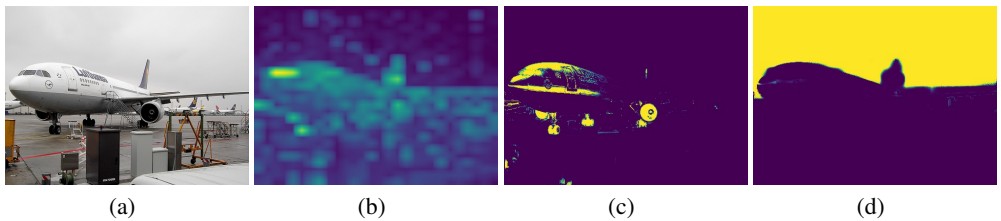

Figure 1: (a) Actual image (b) Hybrid-ViT explainability cue for the "Plane" Class (c) dCRF generated confident foreground (d) dCRF generated confident background (The lighter the color intensity the higher the activation).

Next, we extract pairs of pixels belonging to the same category with high confidence. Additionally, we also consider as neutral, those pixels that were classified by the dCRF as background in the presence of low $\alpha$ and as foreground in the opposite case. Finally, the construction of confident common-instance pairs is now feasible. We consider pairs of positive and negative affinity, in a class-agnostic manner, while we ignore any pair containing neutral pixels. It is worth highlighting that only neighboring pairs are extracted with a radius of 5 pixels. An intuitive figure, showcasing the possible affinities is displayed below.

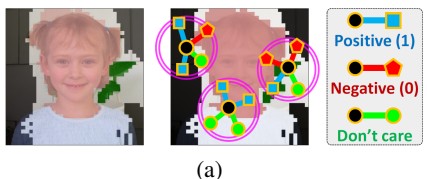

Figure 2: Concept of pixel-to-pixel affinities [image taken from (2)]

### 3.3.2 Training AffinityNet

After having generated the explainability-based affinity targets, we can now train a neural network to generate insignificant $W$ values to those pixels that are semantically unrelated. More specifically,

we utilized the CNN-backbone as trained in the Hybrid-ViT image classification task for feature representation $f^{\text{aff}}$ purposes. In order to adapt to affinity-assignment task, we employed two $1 \times 1$ convolutions on top of the feature map extracted from the Hybrid backbone. The loss used for training the network incorporates three different types of affinities, namely the negative, the foreground positive and background positive affinities. Additionally, we weighted the loss contributions of these three types based on the amount of negative, foreground, and background affinity labels on each training batch. The intuition behind this approach was to avoid only accounting for the most frequent case of background positive relationships due to images containing mostly background content. Based on these the overall loss was computed as :

$$\mathcal{L}_{\text{fg}}^{+} = -\frac{1}{N_{\text{fg}}^{+}} \sum_{i,j} \log(W_{i,j})^{I(i,j \in \mathcal{T}_{\text{fg}}^{+})}$$

$$\mathcal{L}_{\text{bg}}^{+} = -\frac{1}{N_{\text{bg}}^{+}} \sum_{i,j} \log(W_{i,j})^{I(i,j \in \mathcal{T}_{\text{bg}}^{+})}$$

$$\mathcal{L}^{-} = -\frac{1}{N^{-}} \sum_{i,j} \log(W_{i,j})^{I(i,j \in \mathcal{T}^{-})}$$

$$\mathcal{L}^{-} = \mathcal{L}_{\text{fg}}^{+} + \mathcal{L}_{\text{bg}}^{+} + 2\mathcal{L}^{-}$$

with $I$ being the indicator of $i^{\text{th}}$ and $j^{\text{th}}$ pixel sharing the target relationship $\mathcal{T}$. Note that the $\mathcal{L}^{-}$ contributes twice in order avoid unbalance between positive and negative relationships.

### 3.3.3 Refining the Explainability seeds

At this stage, we utilized the predicted pixel-wise affinities to propagate high explainability activations towards the pixels of identical semantic affinity. In more detail, we regarded the predicted affinities as transition probabilities in a random-walk process. By employing this approach, we were able to propagate the highly activated regions based on the semantic relationships predicted from AffinityNet. The transition matrix derives from the predicted affinities as:

$$T_{rw} = D_w^{-1} W^{o\beta}$$

with $D_w$ being a diagonal array applying row-wise normalization to $W$. Additionally, the $o\beta$ operator is applied so that low transitional probabilities are ignored. Naturally, the hyperparameter has to be an integer value larger than one. Next, we compute the expected transitional probabilities of $t + 1$ iterations of the random walk process as:

$$T_{rw} = T_{rw}^{t}$$

Finally, we extract the semantic segmentation masks through refining the explainability seeds $M_c$ for each $c$ class as:

$$\text{vec}(M_c^{\text{new}}) = T_{rw}\text{vec}(M_c)$$

with $\text{vec}(.)$ being the array flatten operator. In our experiments, we used values of 16 and 8 for the hyperparameters $\beta$ and $t$ respectively.

## 4 Experiments

### 4.1 Data

In this project, two different datasets were used: ImageNet (3) (ILSVRC) 2012 along with its mask-annotated ImageNet-Segmentation (4) split and the PASCAL VOC 2012 (17). The ImageNet dataset validation split consists of 1000 object classes with 50.000 images while the mask-annotated split contain 4.276 from 445 classes. The PASCAL VOC, considers 20 image categories with 10.583 and 1450 images in the training and the validation split respectively.

## 4.2 Transformer Explainability

As part of replicating the target paper (1), we conducted perturbation and segmentation tests, while the results are presented in tables 2 and 3 respectively. For the former type of tests, we use a pre-trained ViT-Base network to extract visualizations for the validation set of ImageNet 2012 (3). Afterwards, we gradually mask out the pixels of the input image, from the one with the highest relevance to the one with the lowest when referring to positive perturbation and vice versa in the case of negative perturbation. Consequently, in the first case, we expect to see a high drop in performance when measuring the mean top-1 accuracy of the network while in the second case we expect the overall performance to remain unaffected. Regarding the latter type of tests, we consider each visualization as a soft segmentation of the image and compare it to the ground truth segmentation mask of the ImageNet segmentation dataset[2]. In table 2 we report the AUC metric for the perturbation tests considering the explainability cues corresponding to both the most confident (predicted) and the ground truth class (target). Additionally, in table 3 we evaluate the segmentation quality of the extracted cues by comparing them with the provided ground truth segmentation masks. In Appendix **??** we provide qualitative results corresponding to explainability cues in ImageNet; generated using our ViT-Base implementation.

|  |  | rollout (18) | raw attention | GradCAM (12) | LRP (19) | partial LRP (20) | Target paper (1) | Ours |
|---|---|---|---|---|---|---|---|---|
| Negative | Predicted | 53.1 | 45.55 | 41.52 | 43.49 | 50.49 | **54.16** | 54.13 |
|  | Target | - | - | 42.02 | 43.49 | 50.49 | **55.04** | 55.03 |
| Positive | Predicted | 20.05 | 23.99 | 34.06 | 41.94 | 19.64 | **17.03** | **17.03** |
|  | Target | - | - | 33.56 | 41.93 | 19.64 | **16.04** | 16.38 |

Table 2: Positive and Negative perturbation AUC results (percents) for the predicted and target classes, on the ImageNet (3) validation set. For positive perturbation lower is better, and for negative perturbation higher is better. Table partly extracted from (1).

|  | rollout (18) | raw attention | GradCAM (12) | LRP (19) | partial LRP (20) | Target paper (1) | Ours |
|---|---|---|---|---|---|---|---|
| pixel accuracy | 73.54 | 67.84 | 64.44 | 51.09 | 76.31 | 79.70 | **79.73** |
| mAP | 84.76 | 80.24 | 71.60 | 55.68 | 84.67 | **86.03** | 86.03 |
| mIoU | 55.42 | 46.37 | 40.82 | 32.89 | 57.94 | 61.95 | **62.01** |

Table 3: Segmentation performance on the ImageNet-segmentation (4) dataset (percent). Higher is better. Table partly extracted from (1).

## 4.3 AffinityNet by ViT explainability

For the purpose of generating competent segmentation masks given only image-level supervision, we relied on AffinityNet to refine the initially incomplete explainability cues derived from the Hybrid-ViT image classification network. We evaluated the class-wise mIoU in the PASCAL VOC validation dataset in table 4 where we compare the mIoU performance of the explainability cues prior and post employing the AffinityNet-based refinement (2). In Appendix **??** we provide qualitative results corresponding to the refinement of the ViT-derived explainability cues via the AffinityNet.

## 4.4 Implementation Details

Regarding the replication of paper (1), no training was required as we relied on the available ViT weights pre-trained on the ImageNet dataset. When it comes to utilizing explainability cues derived from ViT architectures for training the AffinityNet, we trained a hybrid-ViT architecture on PascalVoc

---

[2]ImageNet segmentation dataset was obtained from calvin-vision.net.

|          | CAM (10) [VGG-16] | AffinityNet (2) [VGG-16] | Ours [ViT-Hybrid] | Ours AffinityNet [ViT-Hybrid] |
|----------|-------------------|--------------------------|-------------------|-------------------------------|
| mIoU     | 46.60             | **54.00**                | 44.60             | 50.90                         |

Table 4: Segmentation performance on the Pascal VOC segmentation (17) dataset (percent). Higher is better.

while capitalizing on the weights as pretrained on ImageNet. More specifically, we trained for 20 epochs with a learning rate $5e - 3$. The AffinityNet was trained on Pascal VOC training split for 7 epochs with a learning rate of 0.1 using the affinity labels as generated by the ViT explainability cues. In both these training setups, the batch size was set to 8, the weight decay to $1e - 4$ while the SGD optimizer was used. Finally, during training, images were resized to $244 \times 244$ and $448 \times 448$ resolution for ImageNet and Pascal VOC respectively. Moreover, the images were normalized to have 0.5 mean and 0.5 standard deviation for all channels while random horizontal flip and color jittering were employed for data augmentation purposes.

## 5  Conclusions

In the context of this study, we replicated the ViT explainability approach proposed in (1). Additionally, we capitalized on the explainability seeds derived from a Hybrid-ViT architecture to generate competent semantic segmentation labels for weak-supervision. More specifically, the AffinityNet (2) was employed with the purpose of refining the initially incomplete explainability cues into segmentation masks of higher quality. The quantitative results provided in tables 2 and table 3 indicate that we have successfully implemented the explainability method described in (1) since our results are identical to those originally reported in the latter for all the considered metrics. Regarding the AffinityNet, we evaluated the class-wise mIoU performance that we have achieved based on the explainability seeds as generated by the Hybrid-ViT architecture.

Furthermore, according to table 4, we observe that the performance we achieved is lower compared to the one reported in (2), however segmentation masks of improved quality were generated. One reason for that could be the lower quality of ViT explainability seeds compared to the CNN-based ones. Another potential reason for the lacking performance of the AffinityNet, when given explainability cues from ViT architecture, could be that the feature map $f^{\text{aff}}$ in our case, derives from low-level image representation where as in the original paper (2) feature representation from multiple levels were aggregated. Such multi-level aggregation was not feasible in our scenario due to the nature of the transformer architecture.

Concluding, in this work we have demonstrated the feasibility of using ViT-derived explainability cues with the purpose of training the AffinityNet. Although, we were able to increase the quality of the ViT explainability cues by refining them with the AffinityNet, the CNN-based architectures perform better while using lighter models.

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
