# OpenReview forum: "Weakly-Supervised Semantic Segmentation via Transformer Explainability"
_ML_Reproducibility_Challenge/2021/Fall — RC2021_

### Official Review · Reviewer_r7uz · 2022-02-19
**Review of Weakly-Supervised Semantic Segmentation via Transformer Explainability**

**Rating:** 7
**Confidence:** 4

**Review:**

**Scope of reproducibility:** This submission reproduces the work "Transformer interpretability beyond attention visualization” by Chefer et al. (2021). The original work of Chefer et al. proposes a mechanism for propagating relevancy through the layers of a transformer with the objective of interpreting the decisions of a transformer.  In this reproduction submission, the authors successfully reproduce the key experiments of Chefer et al. relating to vision transformers.  The authors then go beyond the work of Chefer et al. by integrating the resulting relevance scores into the pixel affinity propagation framework of Ahn and Kwak (2018). They show that while the pixel affinity propagation framework can be combined with ViT seeds and used to train a Hybrid-ViT architecture, this approach is less effective than the original CNN-based approach of Ahn and Kwak.

**Code:** The authors made use of the code provided in the original works of Chefer et al. (2021) and Ahn and Kwak (2018). However, they replicated substantial parts of the pipelines.

**Communication with original authors:** As far as I can tell, there was no communication with the original authors. However, given that the results of Chefer et al. (2021) were successfully reproduced, it seems likely that this was simply unnecessary.

**Hyperparameter Search:** The authors did not conduct an extensive hyperparameter search for the work of Chefer et al. (2021). However, they instead extended the work to a new application (described below under "results beyond the paper").

**Ablation Study:** The authors did not provide comprehensive ablation studies.

**Discussion on results:** The authors provide a detailed discussion of the reproducibility of the paper, highlighting that most elements of the original work of Chefer et al. (2021) and Ahn and Kwak (2018) were simple to reproduce. They note, however, that it was somewhat unclear how to compute the AUC metrics for the perturbation experiments in Chefer et al. (2021).

**Recommendations for reproducibility:** As noted above, the authors note one component of Chefer et al. (2021) where there is room for improved clarity to enable others to reproduce the work more easily.

**Results beyond the paper:** The authors go considerably beyond the original work of Chefer et al. (2021) by integrating the resulting relevance scores into the Affinity Propagation framework of Ahn and Kwak. Although the experiments do not surpass the state-of-the-art, this experiment shows promising preliminary results.

**Overall organization and clarity:** Overall, the paper is well organised and the writing is clear.

**Extra comments:** I consider this to be a good reproduction study. By providing experiments beyond the original work of Chefer et al. (2021), it offers useful insights for the community.

**References:**
Ahn, J., & Kwak, S. (2018). Learning pixel-level semantic affinity with image-level supervision for weakly supervised semantic segmentation. In Proceedings of the IEEE conference on computer vision and pattern recognition (pp. 4981-4990).
Chefer, Hila, Shir Gur, and Lior Wolf. "Transformer interpretability beyond attention visualization." Proceedings of the IEEE/CVF Conference on Computer Vision and Pattern Recognition. 2021.

---

### Official Review · Reviewer_vkRi · 2022-03-07
**Not a standard reproducibility paper; lack of deeper analyses**

**Rating:** 5
**Confidence:** 4

**Review:**

This paper does not focus on reproducing the results in a paper. It studies how to combine the approaches in two papers [1] and [2]. [2] is about weakly-supervised segmentation, which uses the attention analysis (of the classification networks) results (i.e., attention maps corresponding to the classes) as the pseudo label in the context of weakly supervised segmentation task without label groundtruth. [1] studies the attention analysis of the vision transformer (ViT) based classification networks. The authors use the ViT based method from [1] to replace the CNN based method in [2].

It is a well motivated study for weakly supervised segmentation - trying to use the ViT generated attention maps to replace the CNN generated ones. It is valuable that this paper reports the results observed during experiments.

However, the authors paid few attention to studying the reproducibility of the original results/claims of [1] and [2]. Suppose we are at a viewpoint that the weakly supervised segmentation task can be a surrogate task to evaluate the reproducibility of the attention analysis method in [1] in different contexts. In that case, the paper still lacks discussion in this aspect.

The authors may further analyze why the pseudo label generated by ViT is not good enough for segmentation. This can help to study whether the results/claim of [1] can be reproduced in a different context and may help improve the significance of this work as a technical innovation paper.
[1] H. Chefer, S. Gur, and L. Wolf, “Transformer interpretability beyond attention visualization,” CoRR, vol. abs/2012.09838, 2020.
[2] J. Ahn and S. Kwak, “Learning pixel-level semantic affinity with image-level supervision for weakly supervised semantic segmentation,” in Proceedings of the IEEE Conference on Computer Vision and Pattern Recognition, pp. 4981–4990, 2018.

---

### Official Review · Reviewer_9G6p · 2022-03-10
**Interesting work with limited discussion on hyperparameter tuning**

**Rating:** 8
**Confidence:** 4

**Review:**

The paper implements the work proposed in Chafer et al CVPR 2021 and also incorporates ideas from another work Ahn et al.



1) It would have been better to present a more detailed discussion on hyper-parameter tuning. This is essential considering the work uses an already available implementation.

2)  It will be better to avoid terms like "....mind-blowing..." in a research article.

---

### Meta-Review · Program_Chairs · 2022-04-07

**Recommendation:** Accept
**Confidence:** 4

**Metareview:**

An interesting contribution with relevant results. Some more exploration of hyper-parameter tuning could be a good contribution, and a rephrasing of certain ways of describing results (see reviewer 9G6p's comments).

---

### Decision · Program_Chairs · 2022-04-09

**Decision:**

Accept

**Comment:**

Following the recommendation of reviewers and meta-reviewer, the paper is accepted for ML Reproducibility Challenge 2021, and will be published in the upcoming special edition of ReScience Journal.